# Injury Pathology in Young Gymnasts: A Retrospective Analysis

**DOI:** 10.3390/children10020303

**Published:** 2023-02-04

**Authors:** Emma Williams, Rhodri Lloyd, Sylvia Moeskops, Jason Pedley

**Affiliations:** 1Youth Physical Development Centre, Cardiff School of Sport and Health Sciences, Cardiff Metropolitan University, Cardiff CF23 6XD, UK; 2Sports Performance Research Institute New Zealand, AUT University, Auckland 1010, New Zealand; 3Centre for Sport Science and Human Performance, Waikato Institute of Technology, Waikato 3200, New Zealand

**Keywords:** gymnastics, youth, injury

## Abstract

Gymnastics has a history of high injury incidence rates. However, little is known about the injury pathology in young gymnasts. The purpose of this study was to fill gaps in the literature by providing insight into the injury pathology in gymnasts aged 6 to 17 years. This study was retrospective in design, where injury data were collected using a Qualtrics questionnaire and distributed via social media. The findings revealed that the most common injury site was the lower limb (60.5%), in particular, the ankle/foot (49%) and knee (27%). Overuse injuries and sprains were most prevalent among the lower limb (25% and 18.4%, respectively), and gymnasts seemed to have a tendency to train through injury with modifications to training. In conclusion, lower limb joint sprain and overuse injuries accounted for the majority of injuries in youth gymnasts. These injuries were more frequently reported in girls in the years associated with peak height velocity and beyond.

## 1. Introduction

Training load monitoring, injury surveillance systems and neuromuscular training interventions were demonstrated to be effective at reducing modifiable injury risk factors in non-gymnastics populations [1,2,3,4,5], though this is predicated on existing knowledge of risk factors for injury [6]. O’Kane et al., 2011 [7] revealed that gymnasts aged 10–12 and 13–17 had an increased risk of injury compared with gymnasts aged 7–9 years due to increased growth rates as a result of growth and maturation [8,9,10]. It is also possible that growth and maturation, coupled with gymnastics having a long tradition of early sport specialisation (ESS), pose additional risks to injury, as this encourages those who are successful at a young age to focus solely on gymnastics. ESS was associated with an elevated risk of musculoskeletal injury [11]; however, few studies reported the extent of the injury problem in young gymnasts (<11 years). Previous prospective research conducted on gymnasts aged 7–18 reported that lower limb injuries were the most prevalent (58%), followed by upper limb (21%) and spinal (19%) injuries [12]. Caine et al. 2003 [12] also reported strains to be the most common injury type (31.8%), followed by sprains (19.3%). Previous retrospective research in gymnasts of this age range also reported a large proportion of acute lower limb injuries (59.6%) compared with upper limb (21.6%) and spinal (10.6%) injuries [7].

The majority of injury surveillance and prevention research in gymnastics is conducted among collegiate-level gymnasts [1,2,3,4,5,6,7,11,12,13]. Collegiate gymnasts display higher rates of injuries that require surgery with associated long-term consequences from the increased probability of osteoarthritis [14,15,16]. Nonetheless, younger gymnasts sustain injuries, though the nature of these injuries is likely to be different. Pre-pubertal children are less likely to experience traumatic injuries due to their low body mass and generate lower forces during sport [11]. Despite children being unable to generate high amounts of force, due to their physiology, very young children are at a higher risk of overuse injuries during periods of accelerated growth [1]. Fractures, growth plate and overuse injuries may be more common than severe ligament injuries given the tendency for late-maturing children to take part in gymnastics and their low chronological age [1]. As children experience a growth spurt, muscular imbalances increase alongside increases in height and body mass, causing increased levels of biomechanical loading on lower limb joints. These risk factors put older children at a higher risk of traumatic and overuse injury [2,3]. Previous research also suggested that muscle strains are more prevalent in older adolescent athletes, whereas strains in early adolescence occur near ossification centres [11]. It is important to understand the injury patterns that occur across the whole spectrum of young gymnasts to better understand the nature of the injury problem and to facilitate future work on injury risk reduction practices.

The aim of the current study was to identify injury patterns among young gymnasts aged 6–17 years old. It was hypothesised that younger gymnasts aged 6 to 9 will report a larger proportion of acute injuries, such as fractures and sprains, whereas gymnasts aged 10 to 13 and 14 to 17 will report a larger proportion of muscle strains, overuse and acute severe injuries [2,3,11].

## 2. Materials and Methods

A cross-sectional research design was implemented to assess the injury epidemiology of young gymnasts. An online questionnaire was distributed via social media platforms that asked youth gymnasts to retrospectively recall their injury history for the previous 12 months.

Based on the number of gymnastics memberships within Wales that year (25,000), the sample size was calculated using a confidence level of 90% and a margin of error of 5%; G*Power indicated that a sample of approximately 268 was required to be representative of the gymnastics population in Wales.

Male and female gymnasts aged 6–17 years from all gymnastics disciplines and competitive levels were eligible to complete the survey and be included in the final analysis. Participant responses were excluded if they did not have an injury in the previous 12 months, were not under the age of 18, or did not confirm parental consent via email.

All questions were worded as simply as possible to encourage the children to be the primary respondent; however, adults were made aware in the information sheet prior to this that some questions would require adult supervision to provide more in-depth answers.

The questionnaire (see Appendix A) required the gymnasts to provide information on their age, sex, gymnastics discipline, competition level, training volume per week (hours per week) and sport specialisation (worded to the children as “gymnastics being your main sport that takes up most of your after-school time”). The gymnasts were also retrospectively asked to report any injuries they sustained in the previous 12 months, where the parent/guardian was asked to assist wherever required to help with the accurate recall of details, such as the date of injury, injury location, injury type, mechanism, severity and treatment. The gymnasts were then provided with the opportunity to report up to three injuries in total sustained within the last 12 months.

The participants’ responses were categorised by injury location, type, severity, mechanism, treatment and not applicable (N/A). Injury severity was only reported if the gymnast stated they took time out of the sport; injury treatment referred to whether the gymnasts took time out of the sport to recover, trained through their injury with modifications or trained as normal. If a gymnast did not get diagnosed by a medical professional, they automatically reported a N/A response for injury type. Group comparisons were made between three chronological age categories (6 to 9, 10 to 13 and 14 to 17).

## 3. Results

A total of 249 gymnasts completed the questionnaire; from this sample, 61 respondents sustained at least one injury and completed all the necessary inclusion procedures. The respondents were predominantly female (*n* = 55, 90.2%), and for this reason, only female data were used for further analysis. Some respondents suffered multiple injuries; therefore, a total of 76 injuries were included in the analysis. A total of 17 injured participants were aged 6 to 9, 35 were aged 10 to 13 and 26 were aged 14 to 17 years old. The majority of respondents participated in artistic gymnastics (60.3%), with only 11% of participants engaged in team gymnastics, 9.6% in acrobatics, 8.2% in tumbling, 6.8% in trampolining, 2.7% in rhythmic gymnastics and 1.4% in double mini-trampolining. The competition levels of the gymnasts in this study were national (42%), regional (31%), local/school (18%) and international (9%). Of all the injured participants, 91.2% considered themselves to be “specialised” in gymnastics, and more than 50% of specialised gymnasts did not take part in any other sports. The remaining specialised gymnasts who did take part in another sport stated their extra sports were other forms of gymnastics, such as dancing, cheerleading or lunchtime gymnastics clubs.

### 3.1. Injury Location

In total, 60.5% of injuries occurred in a lower limb, 35.5% in an upper limb and 3.9% in the spine. Forty-nine percent of lower limb injuries occurred at the ankle/foot and 27% at the knee, with the remaining 24% distributed between the shin, thigh, groin and hip. The majority of upper limb injuries were reported in the wrist/hand (46%), the elbow (27%) and the shoulder (23%), with the remaining 4% occurring in other areas of the arm.

When comparing the injury location across age groups, 50% of lower limb injuries and 37% of upper limb injuries occurred in gymnasts aged 10–13 years (Figure 1). Gymnasts aged 10–13 were found to make up half of all ankle injuries and over half (58%) of all knee injuries. In contrast, gymnasts aged 14–17 accounted for half of all shoulder injuries.

Injuries were evenly distributed between the lower limbs (47%) and upper limbs (53%) in 6-to-9-year-olds but were more biased towards lower limbs for both 10-to-13-year-olds (66%) and 14-to-17-year-olds (58%). Spinal injuries were less prevalent across all age categories and none were reported in the 6-to-9-year-old respondents.

### 3.2. Injury Types

Overall, acute injuries accounted for 65% of all injuries while overuse injuries made up 35% of the remaining injuries. Acute injuries were relatively evenly spread between the age groups. However, there were considerably more overuse injuries in gymnasts aged 14 to 17 (48%) in comparison to gymnasts aged 6 to 9 (7%). While the data showed that overuse injuries generally increased in frequency with age, the frequency of acute injuries peaked in gymnasts aged 10–13 years (47%).

#### 3.2.1. Lower Limb Injury Types

Overall, lower limb overuse injuries were the most prevalent, making up a quarter of all diagnosed lower limb injuries, while the second most reported lower limb injury type was sprains (18.4%). Over half (53%) of all lower limb sprains were reported by gymnasts aged 10 to 13 (Figure 2). Lower limb overuse injuries were most prevalent in gymnasts aged 10 to 13 and 14 to 17 (44% and 48% of all lower limb overuse injuries, respectively).

#### 3.2.2. Upper Limb Injury Types

Figure 3 compares the reports of diagnosed upper limb injuries between age categories per injury type. Overall, upper limb fractures (10.5%) and overuse injuries (10.5%) were the most frequently reported injuries sustained to upper limbs. Gymnasts aged 10 to 13 accounted for 50% of upper limb fractures and 40% of sprains. Gymnasts aged 14 to 17 accounted for 62.5% of all upper limb overuse injuries.

### 3.3. Injury Severity and Non-Time-Loss Injuries

Injury severity data were only reported by the gymnasts who took time out of the sport. Overall, gymnasts aged 14 to 17 also reported the most acute injuries with the largest proportion of mild, moderate and severe injuries (Figure 4). Severe acute injuries were reported mainly in gymnasts aged 10 to 13 (43%) and 14 to 17 (43%). Non-time-loss injuries were most prevalent, accounting for 73% of all injuries.

### 3.4. Training Volume and Injury Diagnosis

Gymnasts aged 6 to 9, 10 to 13 and 14 to 17 trained for 11.3, 16.5 and 14.8 h per week on average, respectively, ranging from 1 to 21, 5 to 26 and 5 to 25 h per week, respectively (Figure 5). It was evident that the two youngest age categories in this study were training on average more hours per week than their age (years). Overall, acute injuries accounted for 55.7% of all diagnosed injuries and overuse accounted for 44.3%. Almost half (47%) of all acute injuries were reported among gymnasts aged 10 to 13. Overuse injuries were found to be the highest in gymnasts aged 14 to 17 (48%), followed closely by gymnasts aged 10 to 13 (44%). Gymnasts aged 14 to 17 did not exceed more than 14.8 training hours per week on average. However, gymnasts aged 6 to 9 and 10 to 13 reported 11.3 and 16.5 h per week on average, respectively.

### 3.5. Injury Mechanism

Results indicated that most injuries occurred during a gymnastics skill (54.7%), followed by a landing (20%) and, lastly, falling (9.3%). All but two injuries occurred during training (97.5%), as opposed to during competition (2.5%).

Gymnasts also reported whether they sustained multiple injuries. Overall, 14 gymnasts reported multiple injuries in the previous 12 months (18% of all participants). Four of these gymnasts injured the same limb on multiple occasions, while the remaining ten gymnasts (71%) sustained their second and third injuries in the opposite limb. Gymnasts aged 6 to 9 reported two instances of multiple injuries (12%), gymnasts aged 10–13 reported five instances (14%) and gymnasts aged 14–17 reported seven instances (27%).

### 3.6. Injury Management

Of those gymnasts who reported an injury, 70% continued to train with modifications to their training, 5% were instructed to train through their injury, while the remaining 25% had complete rest and, in turn, reported injury severity. Fifty-one percent of gymnasts who reported training through their injury were aged 10 to 13 years old.

## 4. Discussion

The aim of the current study was to identify injury patterns among young male and female gymnasts aged 6–17 years old. Due to the small sample of males reporting injury data, the findings of this study related to female gymnasts only. The most injured area of the body was the lower limb, making up 60.5% of all injuries. Areas of most concern seemed to be the ankle/foot and the knee. The majority (93%) of overuse injuries also occurred in a lower limb and were reported most frequently in gymnasts aged 10–13. Data indicated that upper limb injuries were most frequent in the wrist/hand (46%). The findings also revealed that 70% of injured gymnasts continued to train after their injury. Overall, very young gymnasts aged 6 to 9 in the current study reported smaller proportions of injury. Gymnasts aged 10–13 seemed to report the largest proportion of injury, where half of all lower limb injuries occurred in this age group. Gymnasts of this age also accounted for half of all ankle injuries and over half of all knee injuries. Gymnasts aged 6 to 9 were less susceptible to severe acute injuries compared with gymnasts aged 10 to 13 and 14 to 17. Overuse injuries were also seen to be the most prevalent in gymnasts aged 14 to 17 (48%) and 10 to 13 (44%).

Results from the current study displayed a similar trend to previous research, with lower limb injuries accounting for 61% of injuries, followed by upper limb injuries (36%) and spinal injuries (4%). Data from this study revealed that lower limb injuries accounted for over half of all injuries across all age categories. Previous literature observed that lower limb injuries accounted for the majority of gymnastics injuries, ranging from 52.9–72.5% in several studies [4,5,6,8,11,13]. A high prevalence of lower limb injuries was reported in the current study and in previous research; this could be explained by the frequency of high impact and repetitive rebounding in gymnastics. These risk factors, coupled with the guidance from the International Gymnastics Federation (FIG) to land with an upright posture, minimal knee flexion and minimal knee and foot separation [9], contribute to increasing the risk of injury during landings in gymnastics. Upper limb injuries seem to be more prevalent in the current study in comparison to previous research. Upper limb injuries previously accounted for 9.2–24.1% of injuries in female gymnasts [4,5,7,8,12]. This could be explained by the inclusion of very young gymnasts aged 6 to 9, which is an age category that was previously under-researched. Gymnasts of this age are expected to attenuate forces up to 3.9 times their body weight at the upper limb on various apparatuses, such as the vault and uneven bars [3]. This repetitive loading on growing structures in the upper extremity can lead to chronic problems, such as physeal stress fractures [3].

Overuse injuries reported in the current study accounted for almost half of all diagnosed injuries; undiagnosed injuries were not considered and were reported as N/A due to the lack of validity. Overuse injuries are defined as an injury caused by repetitive submaximal loading of the musculoskeletal system with inadequate recovery time for subsequent adaptation [1]. Overuse injuries increase with age and were observed to peak around the PHV due to repetitive stress and a rise in training volume on a rapidly growing skeleton [10,17,18]. Early intensive training on underdeveloped and rapidly growing musculoskeletal systems was shown to be a risk factor in young athletes participating in other sports [14,15,19]. The current study demonstrated that the average number of training hours reported by injured gymnasts was 14.1 h per week, where 73% of gymnasts reported training through injury instead of taking time out of the sport. Very young gymnasts aged 10 to 13 years had an average training volume of 16.5 h per week and a maximum of 26 h per week reported, suggesting that gymnast training hours increased during the period that we would typically associate with the pubertal growth spurt. Previous findings also indicated that young gymnasts aged 6 to 13 years old participated in more hours of gymnastics than their chronological age [20]. Furthermore, the volume of training in gymnasts seems to have exceeded recommended guidelines where athletes are advised not to train more hours per week than their age in years since this was demonstrated to increase the risk of overuse injury [17]. The current study revealed that 91.2% of gymnasts considered themselves “specialised“ in gymnastics, which was most likely due to the high training volume, as previously mentioned, and thus, leaving no free time to take part in various sports.

Acute injuries reported in the current study in turn accounted for over half of all diagnosed injuries. The most frequently reported acute lower limb injury was sprains; lower limb injuries, such as anterior cruciate ligament (ACL) injuries, are influenced by the magnitude and timing of force attenuation during landing, and this can be compromised in young gymnasts, as they typically endure high chronic and repetitive workloads during periods of rapid skeletal growth [12,21].

The findings in the current study included very young gymnasts, who are not well captured in previous literature. The current findings suggested that gymnasts aged 6 to 9 experience a larger proportion of acute injuries, such as sprains and fractures, in comparison to overuse injuries, whereas gymnasts aged 10 to 13 and 14 to 17 reported a larger proportion of overuse injuries. Sprains were previously reported to account for 15.2–19.3% of all gymnast injuries and strains were reported to account for 17.7–31.8% [4,5,12] of all injuries. The findings of the current study demonstrated that sprains accounted for 25% of injuries. Over half of all sprains (53%) were reported in gymnasts aged 10 to 13 in the current study. Females of this age typically experience a period of accelerated growth, which may increase the risk of joint injuries due to disturbances to neuromuscular control and a temporal delay in muscular development relative to skeletal growth [22]. Muscular strains only accounted for only 4% of injuries. Muscular strains tend to rise post-PHV. The inclusion of very young gymnasts in this study may be a contributing factor to the lower reports of muscular strains in this data set. Older gymnasts experience a decrease in relative strength and skill deficits that might expose more mature gymnasts to a heightened risk of a musculotendinous injury, such as strains [23].

In agreement with the hypothesis, severe injuries were reported to a greater extent (86% of all severe injuries) in gymnasts after the age of 10. In the current study, reports of injury severity increased with age and training volume. The increase in injury severity was less likely to be associated with injury burden, as overuse injuries tended to be less severe. Increased exposure to gymnastics inevitably exposes gymnasts to a higher risk of injury. Increases in injury severity with age could be caused by external and internal risk factors; increases in height and body mass are generally linked to an increase in age; an increase in age in gymnastics also characterises older gymnasts with more experience with years of training and a higher level of difficulty. Internal risk factors could associate with increased height and body mass, which, in turn, can increase levels of biomechanical loading on lower limb joints, where forces as high as 11 × bodyweight are being attenuated [3]. Injury severity data are under-reported in the previous literature and are possibly invalid due to the definition of injury being used in comparison with the nature of injuries that occur in gymnastics. Injury severity is usually categorised by time loss, which poses issues in capturing minor injuries, whereas, in other sports, the time loss may be reported. In gymnastics, however, training is often modified in a bid to accommodate injuries while enabling ongoing participation; therefore, time loss may not be reported for minor injuries in this population. Non-time-loss injuries are not well captured in the currently available injury epidemiology data. For this reason, Kerr et al. [16] reported novel data on female collegiate gymnasts, stating that two-thirds of injuries seen by athlete trainers did not restrict participation for at least 1 day. This common behaviour displayed by gymnastics coaches to alter training around an injury was reported to increase the risk of severe injury [24].

The mechanisms of reported injuries in this study contradict previous findings; the current study found that over half of the injuries occurred during a skill (54.7%), followed by a landing (20%), compared with two previous studies that recorded a range of 49–80.4% of injuries occurring during the landing [6,7]. Kirialanis et al. [6] carried out a prospective analysis of injuries in 187 artistic gymnasts aged 9–15 years old, reporting a significant increase in injuries occurring during landing on the floor. O’Kane et al. [7] also discovered through retrospective analysis in gymnasts aged 7–16 years old that the landing was the cause of 49% of injuries in 96 female gymnasts. The discrepancies between previous research and the current data might be the result of the optional answers provided in the questionnaire. The gymnasts in the current study were given the multiple-choice answer options of “during skill” and “during a landing”; since most skills and routines in gymnastics involve a landing or rebound, it is possible that the gymnast could not pinpoint exactly when the injury happened.

Almost a fifth of the gymnasts surveyed (18%) reported more than one injury in the previous 12 months. The second injuries were rarely in the same limb (29%), which could be explained by the lack of rest following an injury and, in turn, overusing other body parts. Due to the varied apparatuses used in artistic gymnastics, it is possible that the gymnasts modified their training when working through injuries; if a gymnast removed activities involving the lower limb and replaced all of that volume with an upper-limb-focused apparatus, then there is a rapid rise in training load that the gymnast has not had a gradual introduction to through progressive chronic loading. This “spike” in volume might then be the cause of the injury due to insufficient recovery from the high training loads that they are unaccustomed to. The remaining gymnasts who experienced an injury to the same limb on multiple occasions could be explained by the large proportion of gymnasts who stated that they trained through their injury. For example, an injury to the left leg is going to result in asymmetries during landing, placing greater stress and load on the contralateral limb in an attempt to protect the injury. Previous research suggested that athletes recovering from ACL injuries can experience performance deficits and changes in kinematic and kinetic parameters to the non-injured leg [25]. Although the current study cannot specifically state that this pattern of recurring injury was due to altering training around an injury, the current findings can support this concept, which is discussed in previous literature [16]. Findings from the current study allow for insight into non-time-loss injuries in a very young population, which was not captured well in previous research.

Retrospective injury surveillance approaches were shown to be less reliable than prospective approaches due to recall bias and the under-reporting of injury rates [26,27]. It is possible that injuries were under-reported in the current study due to the voluntary sample approach. However, it was suggested that a fair degree of success is achieved when using a 12-month recall period [28]. Nonetheless, the current study revealed novel data that provide insight into injury patterns in very young gymnasts and highlight gaps in the research, such as injury patterns in young gymnasts and gymnasts who are potentially experiencing periods of accelerated growth. During these growth periods, gymnasts are likely to experience disturbances in neuromuscular control due to increased body mass, longer levers and greater joint torques, which the musculoskeletal system is not prepared for due to delayed strength improvements, as the development of the muscular system lags behind skeletal growth [29,30]. Neuromuscular control of the trunk and lower limb were highlighted as a risk factor for lower extremity injury [17,31,32]. This high prevalence of lower limb injury is of concern for long-term health in gymnasts, as severe knee and ankle injuries are associated with the onset of osteoarthritis and chronic ankle instability [33]. This is important to note, as the manifestation of chronic injuries may affect physical activity in later life, therefore impacting the long-term health of gymnasts.

## 5. Conclusions

Findings from this study revealed that lower limb injuries were common in young female gymnasts and that common injury diagnoses were sprains and overuse injuries. This study also revealed that training volumes were very high in young gymnasts. Injury epidemiology in young gymnasts is also sensitive to chronological age, most likely due to the varying degrees of biological maturity in the experimental sub-groups. Gymnasts reported continuing training with modifications when injured, and very few took time out due to injury. Further prospective research is needed to establish the relationship that training volume and intensity have with injury risk and what internal, modifiable risk factors may exist to inform injury risk in young female gymnasts.

## Figures and Tables

**Figure 1 children-10-00303-f001:**
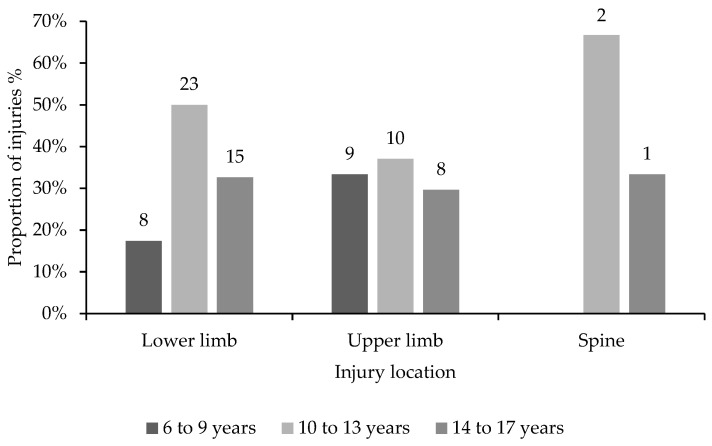
Proportions of lower limb, upper limb and spine injuries distributed across three age categories, with the absolute numbers of lower limb, upper limb and spine injuries in each age category.

**Figure 2 children-10-00303-f002:**
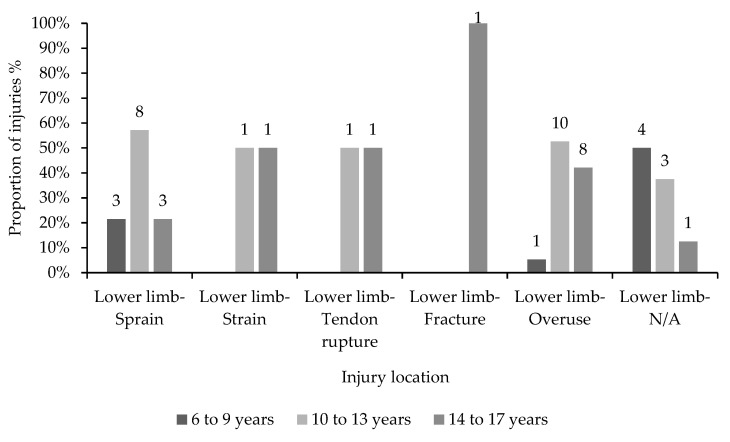
Proportions of diagnosed lower limb injury types distributed across three age categories.

**Figure 3 children-10-00303-f003:**
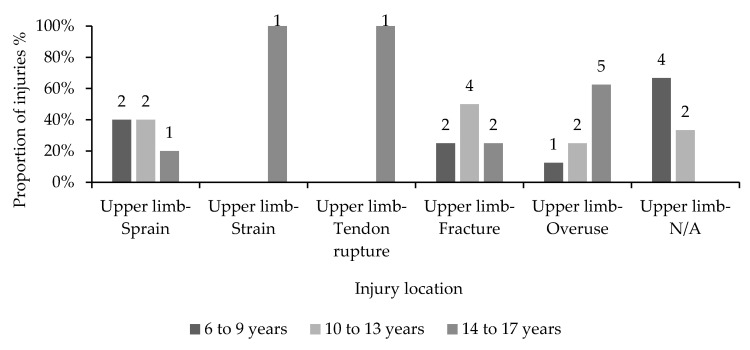
Proportions of diagnosed upper limb injury types distributed across three age categories.

**Figure 4 children-10-00303-f004:**
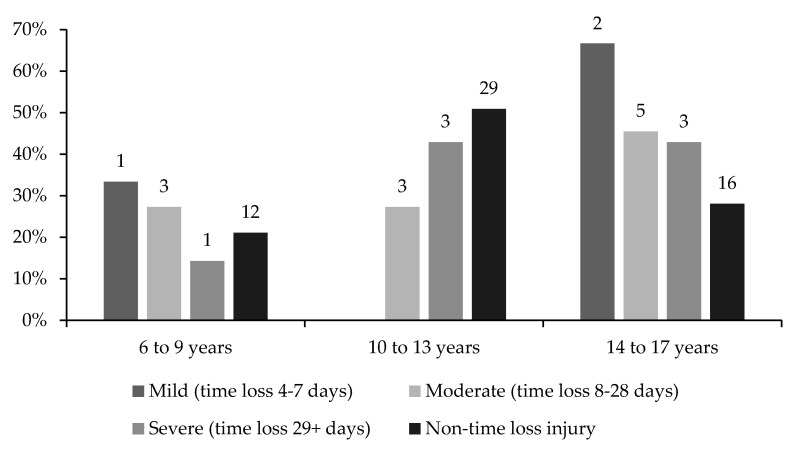
Proportions of mild, moderate, severe and non—time—loss injuries distributed across three age categories.

**Figure 5 children-10-00303-f005:**
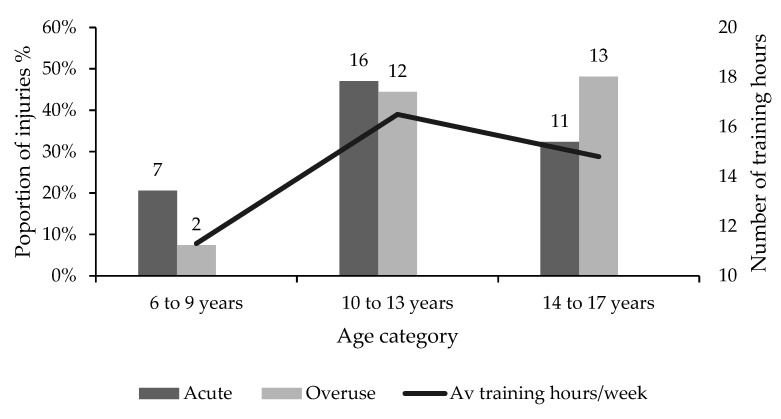
Proportions of acute and severe injuries distributed across three age categories, with the average training hours per week reported by age category.

## Data Availability

Data can be made available upon request.

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
