# Peer review of "Injury Pathology in Young Gymnasts: A Retrospective Analysis"

_children, 2023, doi:10.3390/children10020303_

Round 1

Reviewer 1 Report

dear authors,

It is an interesting topic. My consern is about the consentimet of the parents and the ethical comission. Was this study approved by ethical comission?

The introduction can specify the concerns about this topic in children. The survey questions were verified to be understand for 6 years old and 17 year old children?

best regards

Author Response

Reviewer 1:

It is an interesting topic. My consern is about the consentimet of the parents and the ethical comission. Was this study approved by ethical comission?

Response: Many thanks for your kind words. The journal submission guidelines require a statement to be included regarding ethical approval. This statement is included at the end of the manuscript regarding the process for obtaining informed consent and assent and the approval ethics code from the university institutional review board.

The introduction can specify the concerns about this topic in children. The survey questions were verified to be understand for 6 years old and 17 year old children?

Response: The methods section states parents/guardians were advised they may have to support the child with this questionnaire depending on their age. It is difficult to discern what is meant by “concerns about this topic in children”, the authors assume this is in relation to understanding of the question since injury epidemiology research is frequently conducted in youth. While there are ethical considerations that must be attended to when conducting this type of research, the introduction would be an inappropriate place to discuss these factors. The methods section is more suitable but the journal requires a statement in the subtext at the end of the manuscript to address ethical approval and informed consent, so it seems unnecessary to repeat this information in the methods section.

Reviewer 2 Report

Congratulations to the authors of this article. It is a very interesting article.

It is an interesting article about pathology in young gymnasts. I have some doubts about the article, which I will explain below:

Introduction.

It seems to me a correct introduction making an analysis of the problem. Perhaps I would mention the importance of the different modalities that you cite in the study and the differences that may exist between the different ages that other studies say, if not there is in this sport modality, in others of similar characteristics.

Methodology:

A question due to lack of knowledge, was the minimum number of subjects needed to reach the conclusions reached in the study calculated?

It is necessary to describe in more detail how the data collection was carried out, how the statistics were performed, I would add correlations between ages, gymnastics modalities, injury areas? To know possible correlations.

The hours of training of the respondents were controlled, as well as repetitive injuries....

Results:

In the figures remove the ,0%. If they are all .0% it makes no sense to put it.

The statistics is descriptive we would have to try to give information on correlations, percentages with respect to the population that has been used in this study.

The graphs are not very appealing, perhaps I would modify some of them, so that they are all the same.

Discussion:

More discussion is needed on the subject, since the article talks about many variables about injuries in gymnasts.

Other aspects:

The abstract lacks any relevant results in numerical format that have been taken from this study, as well as the most important conclusions.

As I mentioned before, you include variables such as training hours, level etc... that I think it could be interesting to correlate with some of the results you have seen in this study.

More statistical data should be provided to complete the article, since only descriptive data appear that do not lead to really interesting conclusions.

Author Response

Reviewer 2

Congratulations to the authors of this article. It is a very interesting article.

Response: Many thanks for your kind words and the time taken to review the manuscript.

It is an interesting article about pathology in young gymnasts. I have some doubts about the article, which I will explain below:

Introduction.

It seems to me a correct introduction making an analysis of the problem. Perhaps I would mention the importance of the different modalities that you cite in the study and the differences that may exist between the different ages that other studies say, if not there is in this sport modality, in others of similar characteristics.

Response: The authors are unsure what modalities are being referred to (mechanism of injury, type of gymnastics, type of injury etc). The authors have added lines 33-38 to describe the trends in injury location highlighted in previous literature.

Methodology:

A question due to lack of knowledge, was the minimum number of subjects needed to reach the conclusions reached in the study calculated?

Response: Lines 68-71 have been added to clarify the process for determining required sample size.

It is necessary to describe in more detail how the data collection was carried out, how the statistics were performed, I would add correlations between ages, gymnastics modalities, injury areas? To know possible correlations.

Response: The authors are confused regarding the reviewer’s request; the vast majority of survey questions generated categorical data. It is impossible to perform correlations on anything but continuous data. While the training hours and injury severity did generate continuous data and these two variables might be related and thus worthy of statistical analysis, respondents only reported severity (time-loss from training) if they abstained from training due to their injury. As indicated in the results section and previous literature, gymnasts rarely take complete rest when injured and either train around their injury or train through regardless and so there were insufficient responses reporting severity to be able to perform this statistical analysis. It is common place for epidemiology research to only report descriptive frequencies of injuries as per the two citations below.

Price, RJ., Hawkins, RD, Hulse, MA. And Hodson, A. (2004) The football association medical research programme: an audit of injuries in academy youth football. British Journal of Sports Medicine, 38(4), 466-471.

Read, PJ., Oliver, JL., De Ste Croix, MBA., Myer, GD. and Lloyd, RS. (2017) An audit of injuries in six English professional soccer academies. Journal of Sports Sciences, 36(13), 1542-1548.

The hours of training of the respondents were controlled, as well as repetitive injuries....

Response: The authors are unsure whether this is a question? Respondents reported their training volume in the form of hours of gymnastics per week as indicated in lines 81-82 of the methods section. Training hours were not controlled by the authors, there was no intervention in this study. With respect to “repetitive injuries”, are the authors referring to chronic onset/overuse injuries or reoccurring injuries (being injured more than once)? The phrasing of the questions were framed to try to capture these non-time loss chronic onset injuries while respondents could report up to three injuries which should account for recurring injuries.

Results:

In the figures remove the ,0%. If they are all .0% it makes no sense to put it.

Response: The authors agree with this suggestion and thank the reviewers for highlighting this. The change has been actioned across all appropriate figures.

The statistics is descriptive we would have to try to give information on correlations, percentages with respect to the population that has been used in this study.

Response: As per earlier response, it is not possible to perform correlation on categorical data. Nor can we analyse percentage of respondents who sustained an injury to determine injury rates since the survey requested injured gymnasts to respond. While some non-injured completed the survey, this would not be a representative sample and once they confirmed no injuries had been sustained in the previous 12 months, no more questions were completed.

The graphs are not very appealing, perhaps I would modify some of them, so that they are all the same.

Response: All figures are now uniformly formatted.

Discussion:

More discussion is needed on the subject, since the article talks about many variables about injuries in gymnasts.

Response: The authors are unclear on which “subject” it is that needs more discussion. Lines 198-204 have been added to highlight the trends more clearly in injuries across age groups since we acknowledge that this should be included in the opening paragraph of the discussion as a key theme of the study and thank the reviewer for bringing this to our attention.

Other aspects:

The abstract lacks any relevant results in numerical format that have been taken from this study, as well as the most important conclusions.

Response: Thank you for suggesting these edits to the abstract. Numerical findings have been added to the results section of the abstract (lines 14-16) while the conclusion has been re-written to better reflect the answer to the research question (lines 17-19).

As I mentioned before, you include variables such as training hours, level etc... that I think it could be interesting to correlate with some of the results you have seen in this study.

Response: As per previous responses, correlations are not possible with out dataset. Interactions between training hours and types of injury etc would be interesting, though the likely skew of our sample towards injured participants prevents a valid assessment of the role of training volume on injury since we do not know how many participants performed the same training volume and did not get injured.

More statistical data should be provided to complete the article, since only descriptive data appear that do not lead to really interesting conclusions.

Response: The authors believe that the findings of this study still provide novel insight into the types of injuries sustained by very young gymnasts, especially since the alteration to the code of points introduced by the FIG in 2017.

Round 2

Reviewer 2 Report

Thank you very much for the clarifications received by the authors.

Congratulations for the article. I encourage you to continue researching on this topic.